# Recovery of Banana Waste-Loss from Production and Processing: A Contribution to a Circular Economy

**DOI:** 10.3390/molecules26175282

**Published:** 2021-08-31

**Authors:** Sasha Alzate Acevedo, Álvaro José Díaz Carrillo, Edwin Flórez-López, Carlos David Grande-Tovar

**Affiliations:** 1Programa de Ingeniería Agroindustrial, Facultad de Ingeniería, Universidad del Atlántico, Puerto 081007, Colombia; salzatea@mail.uniatlantico.edu.co (S.A.A.); alvarojosediaz@est.uniatlantico.edu.co (Á.J.D.C.); 2Grupo de Investigación en Química y Biotecnología QUIBIO, Universidad Santiago de Cali, Calle 5 No 62-00, Cali 760035, Colombia; edwin.florez00@usc.edu.co; 3Grupo de Investigación en Fotoquímica y Fotobiología, Universidad del Atlántico, Carrera 30 # 8-49, Puerto Colombia 081007, Colombia

**Keywords:** banana waste-loss, biofuels, circular economy, global banana production, metabolite recovery

## Abstract

Banana is a fruit grown mainly in tropical countries of the world. After harvest, almost 60% of banana biomass is left as waste. Worldwide, about 114.08 million metric tons of banana waste-loss are produced, leading to environmental problems such as the excessive emission of greenhouse gases. These wastes contain a high content of paramount industrial importance, such as cellulose, hemicellulose and natural fibers that various processes can modify, such as bacterial fermentation and anaerobic degradation, to obtain bioplastics, organic fertilizers and biofuels such as ethanol, biogas, hydrogen and biodiesel. In addition, they can be used in wastewater treatment methods by producing low-cost biofilters and obtaining activated carbon from rachis and banana peel. Furthermore, nanometric fibers commonly used in nanotechnology applications and silver nanoparticles useful in therapeutic cancer treatments, can be produced from banana pseudostems. The review aims to demonstrate the contribution of the recovery of banana production waste-loss towards a circular economy that would boost the economy of Latin America and many other countries of emerging economies.

## 1. Introduction

The food sector is divided into different industries, such as the pasta industry, as well as fruits and vegetables, fish, meat and beverages (Figure 1), transforming agricultural raw materials into value-added products [1].

Fruits and vegetables are vital in the daily diet of humans, whose demand is constantly growing due to the increase in population and changes in eating habits. According to the Food and Agriculture Organization of the United Nations (FAO), the world trade of mainly tropical fruits reached a maximum of 7.7 million tons in 2019, an increase of 6.4% (465,000 tons), compared to the previous year [2].

Banana is a tropical fruit grown in more than 130 countries. It is the second most-produced fruit after citrus, contributing around 16% of world fruit production and the fourth most important food crop after rice, wheat and corn. Banana is very nutritious and digests better than many other fruits. Their vast consumption is due to their sensory characteristics and attractive texture and flavor. In addition, it has a high calorie content, with a small amount of fat, and is an excellent source of dietary fiber, vitamin C, vitamin B6 and manganese [3]. The phenols present in banana fruit are the primary antioxidants that provide health benefits. Several phenolics are present in bananas, such as gallic acid, catechin, epicatechin, tannins and anthocyanins [4].

Diets rich in fruits and vegetables with phenols are associated with a lower risk of cancer and heart disease [5]. Phenolic compounds have been used effectively as functional ingredients in foods to prevent lipid oxidation and prevent mold and bacterial growth [6]. Therefore, the banana’s recovery from these secondary metabolites can generate functional ingredients and add value to the banana industry [7].

In 2019, India led banana production with 30.4 million tons, followed by China with 11.6 million tons (Figure 2). The third place was for Indonesia with 7.2 million tons, while Brazil ranked fourth with 6.8 million tons, and the country with the fifth highest production index was Ecuador, with 6.5 million tons produced [8].

According to the United Nations Food and Agriculture Organization (2020) report, banana production had significant growth in the last few years. The market analysis determined that exports reached 20.2 million tons in 2019, due to the growth of banana production in Ecuador and the Philippines, with an estimated growth for 2028 of 135 million tons [9].

According to data from the Ministry of Agriculture and Livestock, total exports of bananas and their products in Ecuador from January to July 2019 were worth two billion dollars, representing 51% of the agricultural production [10].

In 2019, 51,227 hectares of bananas were planted in Colombia. Colombian banana exports were estimated at USD 852.8 million, while those exported to Belgium were worth USD 204.8 million and those to the United Kingdom were worth USD 169.6 million. About USD 149 million-worth were exported to the United States, and USD 124.8 million-worth to Italy [11].

Several countries produced and exported bananas in Europe in 2019, such as the Netherlands, Belgium and Germany, with revenues of USD 790.6 million, USD 782.1 million and USD 242.1 million, respectively [12].

In Asia, banana exports reached a maximum of 5.1 million tons, representing an increase of 42% compared to the 3.5 million tons exported in 2018 [13]. The Philippines has the highest amount of exports of bananas in the Asian continent, and the third place globally (Figure 2). According to preliminary figures provided by the Philippine Statistics Authority (PSA), the export of bananas from this country reached 3.59 million metric tons in 2020, which was 18.35% less than the 4.4 million tons reported in 2019 [14]. Despite a labor shortage, supply chains continue to function despite constraints stemming from measures taken by governments to contain the spread of the pandemic. Sales were affected, however, as was the case with the Philippines [15].

**Figure 2 molecules-26-05282-f002:**
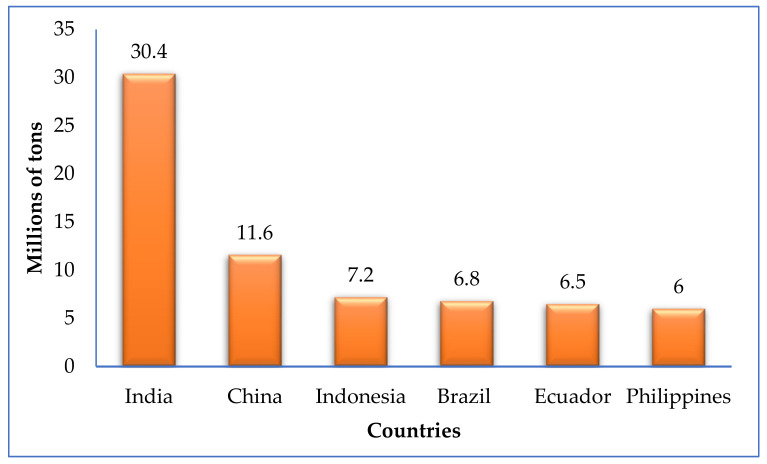
World banana production in 2019 [8,13].

The leading sales from the banana industry derive from fresh banana sales, while the processing rate is less than 20%. Therefore, banana processing has excellent development potential due to the generation and use of processing waste-loss in new production chains. The advanced processing technology is the driving force to improve the added value of commodities and the resource utilization rate of bananas [16].

The banana industry during its production cycle generates large volumes of solid waste-loss derived from maintenance and harvesting processes, highlighting the rachis, pseudostems, leaves and banana peel, which can be used in different processes, such as packaging products and other applications based on bioplastics, thus contributing to the implementation of a circular economy [17].

There are different studies on the processing of bananas and the application of their waste-loss in new production processes [18,19,20,21]. Even so, most of them focus on a single waste, limiting the knowledge and the potential to create new production chains from a natural source. This review offers a comprehensive view of the agro-industrial waste produced in banana processing, potential uses and application in different processes such as biofuel production, wastewater treatment, nanotechnology, bioplastics production as an alternative and an incentive for the development of a circular economy.

The circular economy (CE) can be defined as an economic model based on the efficient use of resources through the minimization of waste, long-term value retention, reduction of primary resources and closed production cycles within the limits of socio-environmental policies [22]. The traditional model is based on a linear consumption where raw materials are extracted and processed into finished products and waste [23]. The circular economy aims to prevent the depletion of resources, close the circuits of energy and materials. The CE implementation will improve sales for companies and quality for consumers while integrating agents and cities, regions and governments in a sustainable and lasting model (Figure 3) [24].

The internal circles of production, reuse and reconditioning demand fewer resources and energy and are more economical than conventional recycling [25]. Climate change is the most imminent environmental problem facing the world. Rising global temperatures have essential effects on human life, food chains, ecosystems and wildlife [26].

An additional benefit of applying a circular economy would help organizations hedge against volatility in commodity prices and rebalance flows of goods, scrap and used products [27]. The Ellen MacArthur Foundation estimates that by 2030 a shift to a CE could reduce resource expenditures in the European Union (UE) by 600 billion euros annually, improving resource productivity by up to 3% per year and generating an annual net profit of 1.8 billion euros [28].

The review aims to demonstrate the contribution of banana production waste-loss towards the implementation of the circular economy by proposing several possible applications (biofuels, water treatment, bioplastic, nanotechnology and organic fertilizers) in order to provide the reader with innovative, sustainable and low-cost ideas that will allow relief in the global economy.

## 2. Banana

Banana is a large-scale, evergreen herbaceous plant belonging to the Zingiberales order, Musaceae family and Musa genus, with high calcium, magnesium and high assimilation of nitrogen emissions [29]. The banana is a fruit with variable qualities, based on size, color and firmness, generally curved and fleshy, covered with green skin, yellow after maturation and brown when ripe. The fruit grows in cones from the plant’s top, up to 15 m between 80 to 180 days [30].

In general, bananas have more than 50 species and dozens of hybrids, large and rhizomatous underground stems, from which their large leaves with powerfully spirally arranged pods start, giving the shape of a false stem (pseudostem, Figure 4).

It belongs to the Musaceae family, order Scitamineae. This family is made up of the *Musa* and *Ensete* gender, monocotyledonous plants, of intraspecific and interspecific crosses between *Musa acuminata Colla* (genome A) and *Musa balbisiana Colla* (genome B) [31]. Currently, banana classification is based on various morphological variations that help in the differentiation of landrace varieties of dessert bananas (AA, AAA, AAB), cooking bananas (AAA, AAB, ABB) and cooking bananas (AAB) [32]. To illustrate this classification more clearly, Table 1 groups together varieties of bananas such as Cavendish or Gros Michel, which usually are produced in countries with a good production index, such as India, the Philippines and Ecuador.

**Figure 4 molecules-26-05282-f004:**
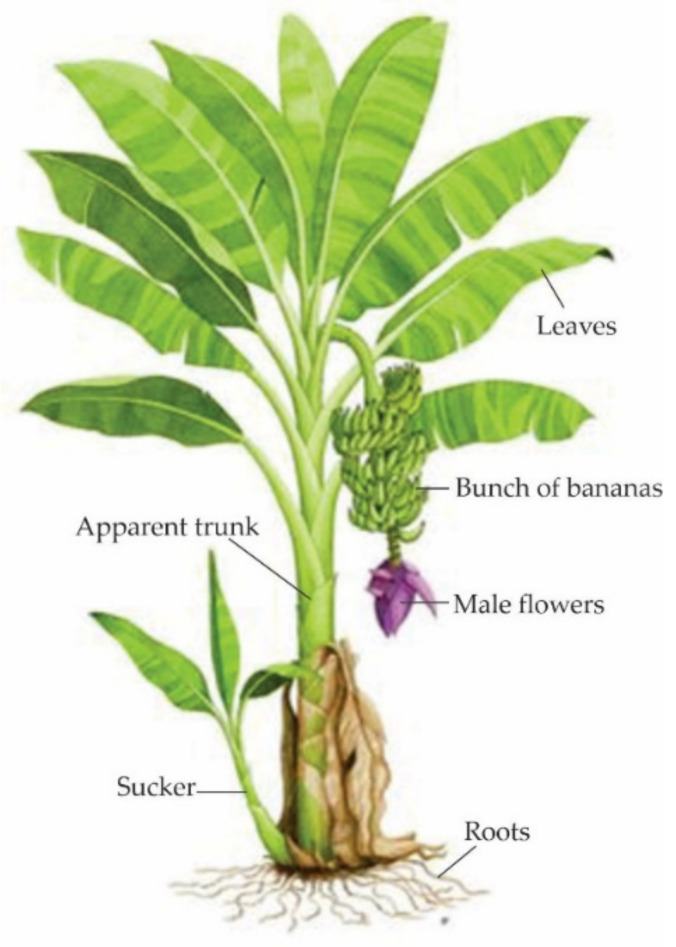
Parts of the banana plant. Adapted from [33].

### Chemical Composition of Banana

Banana contains sugars, acids, vitamin C, amino acids and pectin that form the fruit’s soluble solids content and vary according to the variety grown and the degree of maturity [42]. During the ripening process, ethylene production is induced, which stimulates the production of the enzyme amylase, and breaks down starch into sugar. At the same time, it initiates the synthesis of the enzyme pectinase to act on the pectin between the cells of the pulp, allowing the softening of the plant tissues [43].

The fruit contains various chemical components (Table 2), especially dietary fiber and sugars. During the early part of ripening, sucrose is the predominant sugar, but in the later stages, glucose and fructose predominate [44]. The conversion of starch to sucrose is catalyzed by sucrose phosphate synthase activity, while hydrolysis causes the conversion of starch to non-reducing sugars from sucrose [44]. Bananas are considered a good source of potassium and magnesium, essential components in nutrition and their functional roles involve structural, physiological and metabolic processes in the body [45]. Bananas can provide macro and micro minerals in the diet such as P, K, Ca, Mg, Na, which together with the carotenoids in the plant, play an essential role in the functioning and nutritional value [46].

The peel and pulp are good sources of certain biogenic amines (catecholamines) produced by decarboxylation of amino acids or amination of aldehydes and ketones. Catecholamines include dopamine, serotonin, epinephrine and norepinephrine [47]. Serotonin is crucial for the correct functioning of the nervous system and control of blood pressure in humans. On the other hand, putrescine contributes to physiological processes such as flowering, fruit development and cell division. Putrescine is also used as a quality indicator in various foods [48]. In addition, bananas contain phenolic compounds that help prevent many human disorders such as cardiovascular diseases, obesity and diabetes [49].

**Table 2 molecules-26-05282-t002:** Chemical composition of Banana fruit.

Compound	Content	Unit	References
Carbohydrates	22–88	g/100g DW	[43,50,51]
Dietary fiber	2–5	g/100g DW	[45,51,52]
Protein	1–2	g/100g DW	[43,53,54]
Grease	0.3–1.78	g/100g DW	[52,54,55]
AC	5	mg/100g FW	[45,51,53]
P	350–485	mg/100g FW	[45,46,51]
Mg	26–27	mg/100g FW	[43,45,51]
Vitamin C	12.7	mg/100g FW	[43,51,53]
Vitamin A	12.4	mg/100g *RAE	[51,53]
Folate	20	µg/100g FW	[43,51,53]
Cellulose	5.47	g/100g DW	[52,54]
Hemicellulose	18.83	g/100g DW	[44,54]
Serotonin	28	µg/g DW	[4,46]
Phenols	1–8	g/100g DW	[46,50,54]
Dopamine	7.9–9.9	µg/g DW	[4,56]
Putrescine	25–50	mg/kg DW	[48,57]
Norepinephrine	1.9	µg/g DW	[4]

DW = Dry weight, FW = Fresh weight, *RAE = Retinol equivalent activity.

Bananas, like most fruits, are acidic, with a pulp pH below 4.5. The primary organic acids in bananas are ascorbic, citric, malic and oxalic acid, industrially essential acids because they mainly act as antimicrobial agents, neutralizers and preservatives for food [51].

## 3. Banana Processing

Bananas are considered a household fruit and are mainly characterized by their fresh consumption. However, several finished products are derived from the processing and handling fresh bananas and their components [58]. Banana products include bananas in syrup and dried slices (without frying), frozen bananas, dried bananas, alcoholic beverages and ethanol from bananas, banana powder, jellies, jams, compotes, slices, juices, nectars, drinks, fried slices of banana and banana vinegar, among others [59].

Export bananas are subjected to an intensive quality control process to ensure that they reach the final destination in an optimal state of ripeness and free of stains, dirt and scars from abuse [60]. At the processing plants where the banana is selected and packaged, discarded fruits that have not reached an optimal state of maturity, the right size, have slight bumps or bruises, insect pickets and stains, generate the traditional banana rejection, a waste-loss of the banana processing together with the waste-loss of the post-harvest process [61]. Bananas are classified into three categories according to the Food Codex: Extra Class, in which the bananas must be of superior quality and representative of the variety or commercial type to which they belong; Class I, in which the bananas must be of good quality and exhibit the characteristics of the variety; and Class II, which includes bananas which cannot be classified in the higher classes but satisfy the minimum requirements for fresh consumption [62].

The banana industry is characterized by producing waste-loss from roots, stems, leaves, or any other part of the plant (Figure 5) that is not used during the processing. However, they share the content of lignin, cellulose, pectin and hemicellulose [63].

### 3.1. Peel

Banana peel is one of the essential waste-loss generated from processing. Bananas contain 60% pulp and 40% peel, 7.25 kg of peel produced from a banana box of 18.14 kg [64]. However, the shell contains carbon-rich organic compounds such as cellulose (7.6–9.6%), hemicellulose (6.4–9.4%), pectin (10–21%), lignin (6–12%), chlorophyll pigments and some other low molecular weight compounds (Table 3). If not treated properly, these wastes create an annoying odor due to the natural decomposition and produce gases that contribute to the greenhouse effect [65].

### 3.2. Pseudostems

Pseudostems are the banana stem that supplies nutrients from the soil to the fruits. It is structured by two elements, the nodes and internodes, which make up the floral stem and support the inflorescence in the inner part [69]. The outer part is formed by the sheaths of several bases of rolled leaves emerging from the knots of the corm and the floral stem, arranged in the form of a basal and helical rosette at 120 °C. The size varies between three and five meters high, and its diameter is between 40 and 60 cm [70]. The fiber extracted from the banana’s dried petioles and the pseudostem is utilized in paper [71]. The chemical composition of the dry-based pseudostem is described in Table 4.

### 3.3. Pulp

Banana pulp is a rich source of essential phytonutrients, including phenolic compounds and vitamins (B3, B6, B12, C and E). It also contains carotenoids, flavonoids, amine compounds and dietary fiber [72]. Dietary fibers are polymers of non-digestible carbohydrates that are classified based on their water solubility into two types, soluble fibers (pectin and some hemicellulose) and insoluble fibers (cellulose, lignin and resistant starch) [73]. The chemical composition for banana pulps is described in Table 5.

## 4. Use of Banana Waste-Loss

In recent decades, the world population increased up to 7 billion, generating around 683 million tons of agri-food waste, 34% related to waste-loss of food produced worldwide [76].

The use of this waste-loss has become a topic of great interest, leading to the transition and implementation of a circular economy model as a regenerative economic model [77]. The objective is to close the life cycle of the products through an increase and optimization of use. This objective is also seen in overall favorable balances for the environment and the economy [78].

Sustainable development of the circular economy concept is only possible if technologies are adopted to recover waste-losses. The recovery of food waste-loss opens up new economic growth horizons, allowing the opportunity to transform raw material into a circular loop [79]. The banana industry produces a high amount of lignocellulosic waste that can be used in different recovery processes such as biofuels, wastewater treatment, bioplastics, organic fertilizer and nanotechnology applications. The variety of bananas most used in waste-loss is the *Cavendish* and *Gros Michel* varieties, due to their high availability in different countries such as India, Philippines and Ecuador, detailed in Section 2. In addition, it should be noted that the most used residues are the peel and pseudostem from bananas due to their high content of cellulose, hemicellulose and fiber. *Musa paradisiaca* spp. is one of the most demanded varieties in the harvesting industry. Thanks to the composition of its residues, it can be used in all the applications mentioned above. Banana *Cavendish* is widely used in biofuels because sugars represent the species’ main component of soluble solids. By subjecting them to cellulolytic organisms, a greater quantity of fermented sugars is necessary for biofuel production. Furthermore, the varieties *Williams*, *Musa balbisiana* and *Musa accuminata* have good potential for use in wastewater treatment, bioplastics and nanotechnology applications thanks to their physicochemical properties, such as its lignocellulosic content, its density and its ability to adapt to extreme conditions of climate, soil and water. The chemical composition and physical structure of the waste-loss of banana studied in this paper can be observed in the different tables in Section 3.

### 4.1. Biofuel Production

Due to the rapid development of the automotive industry and environmental pollution problems, the growing demand for petroleum-derived fuel has inspired efforts to explore alternative fuels [73].

The lignocellulosic biomass from the shell, pseudostems and rachis of banana represents a source of promising raw material for ethanol production due to its abundance and high availability. Agri-food waste can be used locally, and does not trigger competition between fuel and food [80]. The production of bioethanol from lignocellulosic biomass (Figure 6) comprises four main stages: the pretreatment of the raw material, the enzymatic saccharification, the fermentation and the recovery of the product—the final crucial step for the process to be economically viable on a commercial scale due to high energy consumption [81].

The banana rachis is chopped to reduce their size to 1 cm in width in the pretreatment, dried and later crushed for their correct use in the other processes [82]. The enzymatic treatment can be carried out in two-liter bioreactors at a concentration of 60% (dry weight) of the shell. The waste-loss are dissolved and stirred in purified water, and during this stage, it is vital to measure the pH, the sugar content and the dissolved oxygen [83]. Fermentation is the anaerobic degradation of organic substances using catalysts presented by some microscopic organisms, such as yeasts. During this process, glucose is transformed into ethanol and carbon dioxide thanks to fermentative processes of yeasts such as *Saccharomyces cerevisiae* [84]. Finally, with the distillation of the alcohol, the aim is to remove the final product due to the difference in boiling point, and as much water is separated to obtain alcoholic beverages with the highest concentration possible (around 40% *v*/*v*) [85].

Finally, with the distillation of the alcohol, the aim is to remove the final product due to the difference in boiling point, and as much water is separated to obtain alcoholic beverages with the highest concentration possible (around 40% *v*/*v*) [85].

On the other hand, the fermentation of banana waste-loss can produce biogas. The production occurs through anaerobic fermentation, one of the most frequent biological processes used to decompose organic materials [86]. Biogas comprises 55–70% methane, 30–45% carbon dioxide and other minor compounds such as water, hydrogen sulfide, ammonia, nitrogen and traces of volatile organic compounds such as siloxanes and terpenes [87]. Biogas quality is highly dependent on the digested substrate(s) and the process parameters. Furthermore, calorific value is conditioned by the concentration of methane in the mixture. For this reason, carbon dioxide and the other impurities must be removed to obtain high purity bio-methane with characteristics superior to natural gas, but with the advantage that the machinery will not be affected by corrosion [88].

Another biofuel alternative that is produced from the use of banana waste-loss is hydrogen. Biomass gasification is a promising technology for converting different raw materials for various energy purposes. This complex thermochemical process converts lignocellulosic materials into a more valuable gas known as synthesis gas through a series of reactions at high temperatures, with the presence of gasifying agents such as air, steam, oxygen or mixtures [89]. The production of hydrogen from ethanol on Cu/Nb_2_O_5_ catalysts promoted with Pd and Ru has received particular attention in research, since it shows promising selectivity for hydrogen selectivity and ethanol conversion in reform reactions using low-cost materials [90]. Hydrogen offers many sustainable end-uses for both small and large scale. Hydrogen end-use options range from highly efficient fuel cells to internal combustion engines. Furthermore, hydrogen can be used as an energy carrier and storage medium in smart grids and other novel applications [91].

Biodiesel is produced from the transesterification of a wide variety of oils and fats (triglycerides), including recycled oils and low-molecular-weight alcohols (mainly methanol and ethanol) [92]. However, biodiesel production from banana waste-loss has been reported mainly at the laboratory level (Table 6). Biodiesel production from the transesterification of triglycerides with short-chain alcohols with the presence of a catalyst consists of three consecutive reversible reactions, where triglycerides are converted to diglycerides, monoglycerides and finally to glycerol. In addition, an ester molecule (biodiesel) formation occurs [93]. Due to the reversibility of the reaction, an excess of alcohol is necessary to improve biodiesel production [92]. Catalysts derived from banana peduncle, peel and pseudostems are some of the solid catalysts based on biomass waste-loss that have been used successfully in recent years (Table 6) [94].

### 4.2. Wastewater Treatment

According to the World Wildlife Organization (WWF), more than 1100 million people do not have access to drinking water worldwide because the growing population demands safe drinking water insurance [101].

Several organic and inorganic products are considered toxic pollutants frequently released into the environment, mainly in surface waters [102]. The industrial and agricultural sectors promote chemical contamination of water due to the various metals, colorants, pesticides, drugs and other compounds that are often used [103].

There is a progressive increase in heavy metals such as chromium, mercury, nickel, cadmium and copper in drinking and wastewater samples that account for the increasing contamination. In the case of hexavalent chromium, it behaves as a toxic element. Chromium is considered harmful for red blood cells, liver, spleen, kidney, soft tissues and bones, affecting the digestive, respiratory and urinary systems [104]. Chromium is a potentially toxic and carcinogenic metal originating from natural processes and anthropogenic activities such as the iron steel, electroplating and leather industries [105]. For removing heavy metals from the water using biomass, techniques such as adsorption, membrane filtration, or chemical precipitations can be used, which are explained schematically in Figure 7 [106].

Banana rachis has been used in adsorption and chemical precipitation methods to remove chromium from wastewater, obtaining a removal efficiency of 99.8% at pH 6.7 [107]. Banana peel has been used as a low-cost adsorption method for oil removal from oil well water, resulting in up to 96% [108]. Electrodialysis (ED) is a membrane technology with many applications, such as water desalination, wastewater treatment and salt production. Additionally, utilizing the bacterial nanocellulose obtained from the shell of Musa spp, filters composed of catalytic membranes can be elaborated [109,110]. Banana peel has been proven to be an excellent low-cost adsorbent, as it is also used in the removal of metals such as nickel and the treatment of synthetic wastewater. Employing electrokinetic remediation, the banana and dioctyl sodium sulfosuccinate shell showed nickel removal percentages of 74.8%. In addition, it is possible to obtain coagulants using banana peel extraction methods to remove turbidity from synthetic wastewater with a removal efficiency of 88% [111,112].

On the other hand, lead, a reasonably malleable corrosion-resistant metal, because it melts quickly, has several uses, and is expected mainly in paints and toxic fumes from industries such as metallurgy, mining and construction industries [113]. It has been shown that through the use of banana peel, removal efficiencies of up to 98% can be achieved with initial concentrations of 100 mg/L, pH 5.0, the adsorbent dosage of 0.55 g and particle size of 75 μm [114].

Another pollutant in wastewater is cadmium, formed by burning metals in the air or pyrolysis of carbonates and nitrates. Cadmium toxicity can affect bones, the respiratory system, kidneys, reproduction and is potentially carcinogenic [115]. Sometimes, the banana marrow comes from the inner part of the pseudostem and is used for livestock, although it is usually discarded without any use [116]. The marrow as a coagulant generated a 98.5% removal of cadmium under acidic conditions due to many functional groups, such as hydroxyl, carboxylic and ether groups [117].

Several studies have been reported on the treatment of wastewater using banana waste-loss (Table 7). Cellulose and hemicellulose from banana pseudostems have created filters to eliminate organic pollutants [118]. In the same way, the rachis and shells of *Musa paradisiaca* have been treated with chemical compounds, such as HCl and NaOH, to obtain activated carbon useful in the chromium removal from wastewater produced in the textile industry [119].

### 4.3. Bioplastics

Polymers can be natural, synthetic or semi-synthetic (Figure 8). Naturals include proteins, enzymes and polysaccharides, while synthetic plastics are thermoplastics, moldable with heating, or thermosets, which are not moldable under heating [122].

Bioplastics are also produced using other polysaccharides such as polyhydroxyalkanoates, collagen and lipids from vegetable oils. In this sense, food waste-loss as an initial raw material is a good option for elaborating biodegradable materials [123]. Starch is a natural polymer of glucose. It is considered a mixture of two polysaccharides: amylose and amylopectin. Starch constitutes the reserve of plant nutrients, and behind cellulose, it is the most abundant carbohydrate in nature [124]. Starch is not plastic, but it can become plastic through polymer technology or fermentation using various techniques such as casting, mixing, extrusion, or injection molding. Commercially, 50% of bioplastics are prepared from different starches [125].

**Figure 8 molecules-26-05282-f008:**
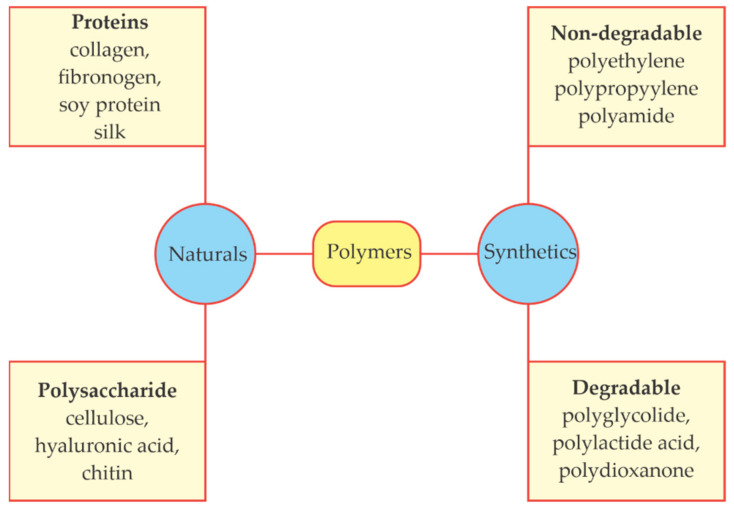
Examples of biodegradable and non-biodegradable natural and synthetic polymers. Adapted from [126].

The peels must be properly washed, cut and then heated with water to produce bioplastics from banana starch. Then they must be crushed and mixed with compounds such as acetic acid, glycerol, or sodium hydroxide [127]. The mixture is subjected to heating to 220 °C with constant stirring, and is allowed to dry at room temperature for one day, followed by heating again in an oven at 85 °C for two hours [128]. Compounds such as glycerol, sodium hydroxide and hydrochloric acid are essential in the bioplastic manufacturing process as they improve the plasticizing properties of the final product [129].

Polyhydroxybutyrate (PHB) is a biodegradable thermoplastic synthesized by submerged fermentation from renewable raw materials. It is a polyester composed of highly crystalline, rigid and brittle linear elastomers with chemical resistance to hydrolysis [130]. PHB has a methyl functional group and an ester linking group that provides the characteristics mentioned above [131]. More than 20 bacterial species include *Bacillus megaterium*, *Methylobacterium rhodesianum*, *Alcaligenes eutrophus*, *Methylorubrum extorquens*, *Pseudomonas putida*, *Sphaerotilus natans* and *Escherichia coli*, among others, are known as producers of PHB [132]. Through sequential steam pretreatment, enzymatic hydrolysates can be produced from the banana pseudostem, reducing PHB production costs and improving the commercialization [133]. Due to the biodegradability, biocompatibility, availability and physical properties comparable to petroleum-based thermoplastics, PHB is a potential substitute in the biomedical and packaging fields [134].

On the other hand, film food packaging avoids the loss of water, which causes a modification in texture and rigidity. In addition, food-packaging delays chemical changes such as color, aroma and nutritional value, since they act as a barrier against gases or volatile loss compounds that influences chemical and microbiological stability and avoid food decay and loss [135]. Banana leaves are used to make packaging (Table 8) due to their availability and ease of processability. Furthermore, leaves are large-size materials that have a degradation time of approximately 28 days after use. In addition, leaves can be stored for six months under proper conditions [136]. Biofilms made from starch share the characteristics of petroleum derivatives, while possessing advantages such as biodegradability and low toxicity [137].

### 4.4. Nanotechnology

Nanotechnology is a field of vertiginous development in recent years, thanks to the multiple applications in medicine, food, cosmetics and pharmaceuticals, among other fields [143]. The main advantage of nanometric organic substances is their higher ingestion-absorption and better bioavailability in the body, compared to ordinary counterparts on a micro or macroscopic scale [144]. In this sense, lignocellulosic materials are a widely used green alternative to produce materials on a nanometric scale [145]. The smart release of nutrients, the bioseparation of proteins, the rapid sampling of chemical and biological contaminants, the intelligent packaging and the nanoencapsulation of nutraceuticals are emerging nanotechnology topics in the agri-food industry [146].

Some uses of *Musaceae* waste-losses have been reported, especially cellulose production (Figure 9), while others in the field of materials such as the production of binder-free boards and carbonaceous preforms [147].

#### 4.4.1. Cellulose and Nanocellulose

Cellulose production begins with pseudostems cutting and grounding to reduce their size for the subsequent stages. After that, a chemical analysis is applied to determine the composition of the samples [148]. Then, an alkaline treatment with NaOH at 15% *w*/*v* for 100 min under continuous stirring to liberate the cellulose fibers. Cellulose bleaching consists of removing residual lignin using 0.5% NaClO for one hour at 70 °C. Finally, the material is washed and dried at 80 °C [149].

This biopolymer has interesting mechanical properties, including low density, high rigidity, high strength, biodegradability, thermostability and low thermal expansion [150]. Combinations of chemical and mechanical processes have used raw materials to remove non-cellulosic compounds and reduce size [151].

When cellulose has been obtained, it is subjected to shrinking methods to obtain fibers at the nanometric scale, nanocellulose. In this respect, it is claimed that cellulose fibers are produced by subjecting banana residues to various chemical processes, while nanocellulose is the transformation of cellulose at the nanoscale [152]. Nanocellulosic materials can be obtained in cellulose nanofibers (CNFs) or cellulose nanocrystals (CNCs), depending on the extraction procedure. In this process, banana peel and pseudo-stems biomass are used as the direct raw material for the production of nanocellulose [153]. It is possible to find cellulose nanofibrils (CNF) to produce biobased materials. Among its properties are crystallinity, high capacity to form films/nano papers with low thermal expansion, high optical transparency, tensile strength, Young’s modulus and barrier (to oil, oxygen and water vapor), in addition to abundancy and non-toxicity [154].

Another alternative to produce nanocellulose is through the use of microorganisms such as *Gluconacetobacter xylinus* or *Medusomyces gisevii*. Banana residues are used as a food source for the bacteria that produce bacterial nanocellulose (BNC) when the organic substrate (sugar, fructose, glycerol) is polymerized during cultivation [155,156].

Using the bacterium *Gluconacetobacter xylinus* in fermentation to produce water filter membranes, the synthesis of bacterial nanocellulose (BNC) from *Musa paradisiaca* (Table 9) was achieved under a pH 4.0, urea at 0.5% by weight and variable sucrose contents (5%, 10% and 15% (*w*/*v*) [157]. Additionally, using rotten banana together with *Komagataeibacter medellinensis* and 26.4 g/L of glucose, 16 g of BNC was obtained after 12 days of fermentation in a plastic bioreactor in Colombia [158]. BNC appears in a twisted band with diameters ranging between 20 nm and 100 nm and a micrometer length. Although BNC has the same chemical composition as other types of nanocellulose, BNC exhibits higher purity, water holding capacity and crystallinity, which leads to excellent thermal and mechanical strength [159]. The property of nanocellulose to form liquid crystals is used in various applications, since they retain order when dried. They have been used to make anti-counterfeiting devices, printing inks and iridescent materials [160], and also have applications in the food industry to prepare creams, glazes, sauces, stabilizing oils and fats [161].

Banana residues are not only used to feed microorganisms that produce BNC. This mechanism is also used in various recovery processes, such as obtaining biofuels and producing enzymes. For example, it has been proven that using banana peel as an energy source for *Saccharomyces cerevisiae* and *Cryptococcus* sp. makes it possible to produce oleaginous and indigenous yeast with applications in ethanol production and biodiesel, respectively [96,162]. It has been proven that using *Saccharomyces* sp. and fermented sugar cane inulase can be synthesized using banana peel as substrate. Furthermore, using banana residues as a carbon source for *Bacillus subtilis*, it is possible to obtain alpha-amylase [163].

#### 4.4.2. Silver Nanoparticles

Commercially available silver nanoparticles are synthesized through wet and dry synthesis procedures and approved in consumer products. Many methods have been reported to synthesize silver nanoparticles using chemical, physical, photochemical, green and biological routes [164]. The chemicals used in the chemical synthesis processes are very reactive and can have potentially harmful effects on the handler and the environment and the high cost of production is another factor. This has led to an increase in the focus on green synthesis methods [165]. In green synthesis, the mechanism involved in the production of silver nanoparticles (AgNPs) consists of three steps: reduction, growth and stabilization of silver nanoparticles (Figure 10) [166].

Due to their reductive properties, banana leaves can be used in the organic synthesis of silver nanoparticles. The reducing sugars glucose, fructose and xylose allow Ag+ from the AgNO_3_ to be reduced and form Ag0, which is stabilized by starch or some other polysaccharides shown in Figure 10. Leaves are pulverized and mixed with AgNO_3_, HCl, NaOH and FeCl_3_, for a subsequent Soxhlet-type extraction, then filtered to remove impurities and stored in a refrigerator [167]. Interestingly, some polysaccharides have been used for silver nanoparticle synthesis. For example, chitosan, one of the essential polysaccharides, can be used in AgNPs green synthesis as a reducing agent and stabilizing agent. Starch is also used as a capping agent and β-D-glucose as a reducing agent. Moreover, cellulose and its derivatives and components can reduce and stabilize agents in AgNPs synthesis procedures [168].

Silver nanoparticles have great utility due to their different properties, thanks to their size, optical, magnetic and electrical shape, usually used for coating materials, biosensors, antimicrobial applications, electronic components and composite fibers [169].

#### 4.4.3. Nano Fertilizers

The vital interests of the use of nanotechnology in agriculture include specific applications, such as the development of nano fertilizers or nano pesticides to promote the levels of nutrients and increase the productivity of the crop with the decontamination of soils, waters and protection against various insect pests and microbial diseases [170]. Banana peels are crushed and mixed with potassium hydroxide; then, the mixture is filtered and heated up to 70 °C, with continuous stirring at 300 pm for organic fertilizer production [171]. In addition, *Williams* banana peel has been used to obtain nano fertilizers, obtaining sizes in the range of 19 to 59 nm (Table 9) [172]. Nano fertilizers improve plant metabolism and nutrient absorption through nanometric pores, facilitated by molecular transporters or nanostructure cuticle pores. An average efficiency gain with the application of nano fertilizers has been verified between 18–29%, compared to conventional fertilizers [173].

**Table 9 molecules-26-05282-t009:** Studies on the production and application of nanomaterials from banana processing waste-loss.

Waste	Nanomaterial	Product or Application	Study	Conclusions	References
*Musa* *Balbisiana* leaves	Silver nanoparticles	Microwave-assisted biosynthesis	Laboratory-scale study. Ten milliliters of banana leaf extract were used in 40 mL of known concentration (0.1 mM) of an aqueous solution of AgNO_3_. It was then heated in a microwave to 160 °C and allowed to dry for one day.	The particle size was in the range of 80–100 nm, and they are crystalline. Nanoparticles were used effectively in anti-cancer study activities.	[167]
		Green synthesis	The laboratory-scale study used a 50 mL filtered extract of banana leaves with 1.0 M NaOH and 10 mL silver nitrate. The mixture was left in a microwave at 600 W for five minutes.	The silver nanoparticles synthesized by the green method exhibited an absorption maximum at 410 nm. Nanoparticle micrographs indicated spherical silver nanoparticles with a size range of 20 to 300 nm.	[164]
*Musa* *Paradisiaca* shells	Bacterial Nanocellulose (BNC)	Flat plate for desalination	Laboratory-scale study. The membrane was prepared in three compositions: T1 (BNC 60%, micro cellulose 20% and silica 20%), T2 (BNC 50%, micro cellulose 20% and silica 30%) and T3 (BNC 40%, micro cellulose 20 % and silica 40%).	The T3 membrane had the highest maximum flux value of 4.41 × 10^3^ L m^−2^ h. The T2 membrane had the highest desalination value of 4.89%.	[159]
		Filter membrane	Laboratory-scale study where BNC synthesis was successfully achieved using the bacterium *Gluconacetobacter xylinus* in a fermentation process under pH 4.0, 0.5% urea and variable sucrose, 5%, 10% and 15% (*w*/*v*).	The nano-cellular nanofiber produced from banana peels had diameter sizes between 30 and 50 nm applied in water filter membranes.	[157]
Banana *Williams* shells	Nanoparticles extracted from the shell	Nano fertilizer	Laboratory-scale study where the shells were crushed and mixed with potassium hydroxide; then, the mixture was filtered and heated up to 70 °C, stirring continuously at 300 pm. Urea and citric acid were added dropwise until pH 5.0.	The size of the nano-fertilizer constituents ranged from 19 to 55 nm. Nanoparticle sizes of 40 nm and 55 nm were obtained.	[171]
*Musa* spp. Shells and pseudostem	Nanosilica	Composite polymer	Laboratory-scale study. The samples were made by pouring a mixture into an open mold. Curing was carried out at room temperature under pressure. Banana fibers (5 wt. %) were used with 0.1 wt.% of nano-silica fillers.	The polymer had a density between 0.8–1.5 g/cm^3^ and a hardness of 50–92 on the Rockwell scale.	[174]
*Musa* spp. husk, canola straw and buffalo manure	Zero Valent Nano iron (nZVI)	Ecological synthesis	Laboratory-scale study. Shell extract was used to reduce Fe ions. Five milliliters of filtered extract were mixed with 5 mL of freshly prepared 1.0–5.0 mM aqueous solution of FeSO_4_ with constant stirring at room temperature.	The formation of nanoparticles could be observed by UV-Visible spectroscopy at a wavelength of 150–550 nm. The optimal concentration for the synthesis of nZVI was 1.0 mM Fe^2+^ ions.	[175]
	Fe_3_O_4_ nanoparticles	Methane production	Laboratory-scale study. Fe_3_O_4_ was added in five different doses (0.4, 0.5, 0.81, 1.22 and 1.63 mg) in two proportions of straw: manure (40:60) and waste-loss of banana: manure (60:40), based on 5 g of volatile solids (VS).	A methane yield of 256.0 mLCH_4_/gVS and 202.3 mLCH_4_/gVS was obtained for straw: manure and banana: manure, respectively.	[176]
*Musa* spp. pseudo stems	Nanocellulose fiber (FNC)	Soluble packing material	Laboratory-scale study. Nanocellulose was used together with polyvinyl alcohol (PVOH). The PVOH was added in a Baker with thermostat at 80 °C and FNC was added in different amounts (1, 2 and 5 g). Then, ultrasound treatment for one hour.	The solubility of the package was 94.57%. The tension strength was 2.36 kg f and Young’s model of 59.16 N·mm^−2^.	[177]

### 4.5. Organic Fertilizers

Organic fertilizers can be defined as a product of the natural decomposition of organic matter by fermentation processes, which varies according to the type of fertilizer to be prepared. The degradation occurs naturally through the air, the sun, microorganisms and water [178]. Organic fertilizers come from composting organic compounds mediated by microorganisms, improving soil quality and providing nutrients to crops, thus reducing chemical fertilizer use and the undesirable polluting factors production [179,180]. Composting systems can be classified depending on the level of aeration in the open-air system, semi-open system and confined system, where the latter is ideal for controlling aeration [181]. Banana peels are rich in micronutrients, suggesting their use to improve soil quality and crop yields, either through their use as an organic soil conditioner or to produce compost to meet specific plant requirements [182]. To obtain the cake, the remains of banana leaves and pseudostems are subjected to endogenous hydrolysis, extraction and neutralization processes. Subsequently, aerobic degradation leads to compost [183].

Composting is done by the layered pile method (Figure 11), with an approximate volume of 1 m^3^. Banana waste-loss is cut and mixed with organic matter for its subsequent degradation process. Water is sprinkled over compost piles to keep the mix moist, with approximately 50% moisture content [184]. The process takes about twelve weeks when the core temperature of the material is close to the air temperature. During this period, the composting material is mixed at least once a week to avoid excessive temperatures from the beginning of the process [185]. The application of this type of fertilizer improves the structure of the soils allowing better absorption of nutrients such as N, P and K by the plants [186]. Compost eliminates harmful bacteria and fungi, neutralizes pH, improves soil texture and improves water retention. Soils with applied compost adsorb better nutrients that benefit plants, thus creating healthier and nutrient-rich foods [187].

Bokashi is one of the most used organic fertilizers for the improvement and fertility of soils. The production process is short and includes many materials such as chicken manure, black earth, ash or coal. Bokashi’s color is grayish, with sand apparency [188]. Before mixing the ingredients, the waste-loss (including peels and pseudostems) are chopped until obtaining particles of approximately 2 to 5 cm. After that, particles are scattered and left to dry in the shade for three days. The earth is sieved to exclude any foreign material such as stones, sticks, or other foreign materials [189]. Molasses is diluted with water, just like bread yeast. The materials are grouped in layers to effect mixing, forming a trapezoidal mound or pile. Subsequently, they are turned to homogenize the components until they form a mound again with a height of approximately 90 cm [190].

The bokashi can be obtained easily and quickly. The soil application also favors plants and is an excellent source of nutrients for vegetation. In addition, it improves water uptake and increases resistance against erosion [191]. This fertilizer constitutes a sustainable alternative with a profitable production that improves crop productivity, maintains good soil quality and efficiently manages available resources [192].

Vermicomposting (VC) involves the biodegradation of organic matter by worms to maintain the flow of nutrients from one system to another [193]. There are several studies on the use of vermicomposting to produce organic fertilizers (Table 10).

Decomposition begins in the gizzards of earthworms, where organic matter is digested. Earthworms can accumulate heavy metals, contaminating organisms and increasing microorganisms’ resource availability by crushing the waste into smaller particles [194]. Organic waste, such as the peel, leaves and pseudostems from bananas, is crushed and spread in layers exposed to sunlight for five to ten days to eliminate pathogenic microorganisms and harmful gases [195]. The banana leaves are valuable organic material because worms can transform them into humus, allowing nutrient transfer to the soil [196]. Among essential parameters for vermicomposting, humidity and temperature are controlled by spraying water on the bed, keeping the temperature at 35 °C and humidity between 50 and 60%. The solid waste is weighed and placed in a container, then sprayed with water to hydrate it to within 50–65% humidity. After that, earthworms are introduced into the solid waste for the decomposition process [197]. The mixture is then covered to protect the worms from sunlight [198].

When used as a fertilizer, VC positively impacts soil quality, plant growth, crop yield and nutritional value. The use of VC in the soil also improves physicochemical characteristics (aggregation, stability, pH, apparent density and water retention capacity) [199]. Likewise, vermicomposting contributes nutrients to the soil such as nitrogen, phosphorus and potassium, allowing an increase in the productivity of crops [200].

**Table 10 molecules-26-05282-t010:** Studies on the production of organic fertilizers from banana waste-loss.

Waste	Technique	Study	Conclusions	References
Pine sawdust and peels of mango and *Musa* spp.	Bokashi	The beds or piles were prepared in a greenhouse. The ingredients were incorporated in the following vertical order: 0.2 m of organic vegetable waste, 0.1 m of manure, a sheet of lime, 0.1 m of forest soil, charcoal, and again 0.2 m of plant waste-loss.	It was observed that the pH ranged from 7.9 to 8.4, which corresponds to modern alkaline. Fermentation lasted 60 days with a maximum of 30.6% organic matter.	[192]
		The waste-loss were mixed in a 1:1:1 ratio and the mixture was homogenized. The piles were turned daily for the first ten days to maintain the temperature of 70 °C. The decomposition process lasted 21 days.	The Bokashi presented electrical conductivity of 8.97 mhos/cm, the potassium content of 4.3 mg/kg of bokashi and sodium content of 161.0 mg/kg, thanks to the Bokashi content of the banana pulp.	[201]
Fruit peels, Pseudo stems and leaves of *Musa* *Paradisiaca*	Vermicomposting	The shredded waste was spread in layers and exposed to sunlight for ten days; then, they were doused with water to hydrate them. The earthworm *Eudrilus eugeniae* was introduced into the solid waste-loss ratio of 5:1 (g/earthworm).	The best result in the chemical composition of the prepared vermicompost was 17.21% of nitrogen, 10.24% of phosphorus, 48.32% of potassium and a carbon-bond-nitrogen ratio of 29, promising for its application in crops.	[195]
		Banana pseudo stems were enriched with cow manure in different proportions using *Eisenia fetida*. A humidity level of 60% was maintained throughout the process.	The chemical analysis showed a gradual increase in plant nutrients such as P, Ca, K, Mg and Fe. The total transformation of the raw material lasted 60 days.	[198]
		Banana waste-loss and mango litter were mixed separately with dry and powdered cow manure (40% cow manure and 60% organic waste) for 30 days under a shady place. Then, *Eudrilus eugeniae* and *Eisenia foetida* were added to finish the process.	Banana waste was a better substrate than mango litter in terms of time required for composting, amount of compost produced and conversion percentage.	[202]
Municipal waste, peel and pulp of *Musa* spp.	Compost	In total, nine containers (170 L) were filled, containing 50 kg of solid waste. Microorganisms were inoculated together with cow manure and waste-loss were cut into small 2 cm pieces. The leachate was made through holes in the bottom of the containers.	The final product had nutrient values of N—2.13%, P—0.57%, K—7.68%, Ca—16,000 mg/kg, Mg—14,600 mg/kg, Iron—113 mg/Kg, Cu—89 mg/kg and Zn—154 mg/kg, as well as lower concentration of heavy metals.	[203]
		*Hermetia illucens* was used with a banana peel pre-treatment to improve the composting process. The shells were homogenized in a blender and subsequently mixed with ethanol, methanol, chloroform and nitrogen.	The mixture containing nitrogen produced the highest final weight of larvae (134 mg). Pre-treatment increases the waste to biomass conversion ratio.	[204]

## 5. Conclusions

The banana industry generates a large amount of waste-loss that can be used in various recovery processes, such as biofuels, wastewater treatment and the production of bioplastics focused on implementing a circular economy. Due to its high content of organic compounds rich in carbon such as polysaccharides, banana peel has been used mainly to obtain bioplastics with high degradation rates, and produce biofuels such as diesel and ethanol. The leaves from the banana plant are used to produce biodegradable packaging and utensils and organic fertilizers. Due to their fibrous composition, the banana’s pseudostems are used to obtain bioplastics requiring little time for biodegradation. Cellulose and hemicellulose banana waste-loss content is of great importance in nanotechnological processes, especially in producing green nanoparticles. It has been shown that the leaves, rachis, pseudostems and banana peel have a high potential for use, which can be reused in various recovery processes. In addition, it allows to close production cycles and reduce the accumulation of waste from the banana industry through its use in different applications, contributing to the growth of the circular economy.

Despite the wide range of application possibilities for banana industry waste, production techniques must be improved to obtain products with better characteristics, such as improving the biofuel process’s pre-treatment technique to produce more energy-efficient end products. However, the lack of pilot-scale studies and the industrial implementation of these wastes is evident, suggesting the need for more studies at higher-scale applications. We cannot forget that companies seek to innovate their productive activities to obtain products with added value every day. For that reason, it is necessary to implement the various possibilities exhibited in this review to add value to the banana industry in a sustainable way, contributing to a circular economy.

## Figures and Tables

**Figure 1 molecules-26-05282-f001:**
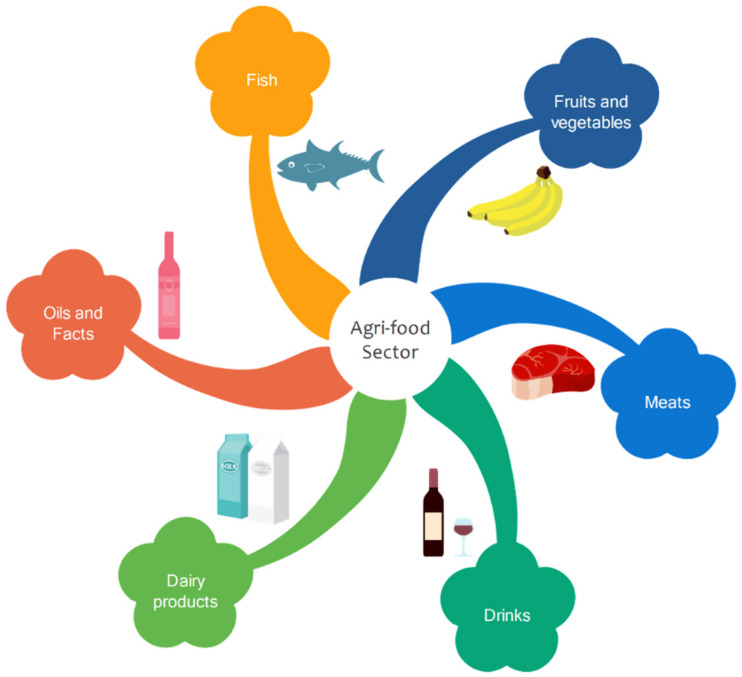
Different industries in the agri-food sector.

**Figure 3 molecules-26-05282-f003:**
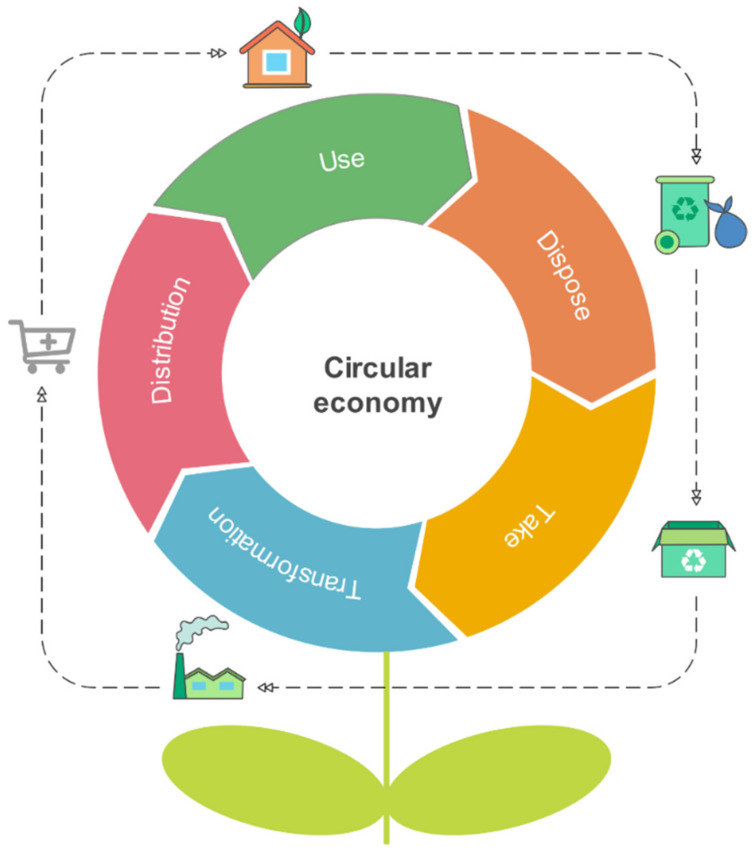
Scheme for a circular economy cycle.

**Figure 5 molecules-26-05282-f005:**
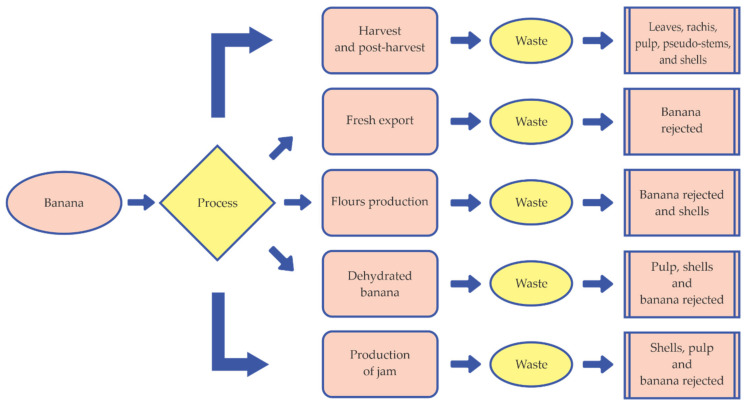
Scheme of production of waste-loss from the banana processing.

**Figure 6 molecules-26-05282-f006:**
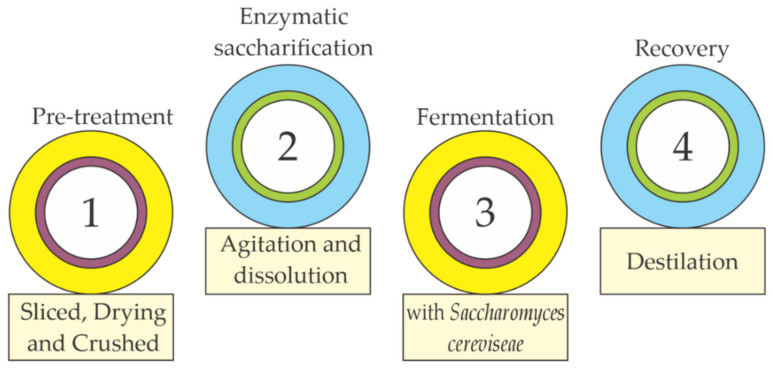
Stages of the bioethanol production process. Adapted from [81].

**Figure 7 molecules-26-05282-f007:**
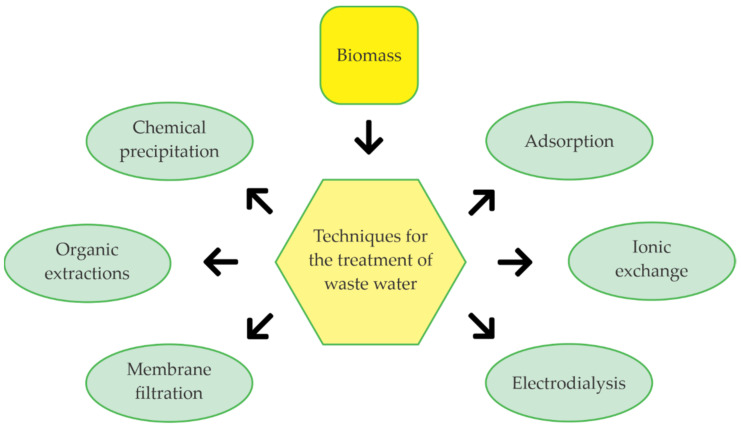
Techniques for the removal of heavy metals present in the water through banana biomass.

**Figure 9 molecules-26-05282-f009:**
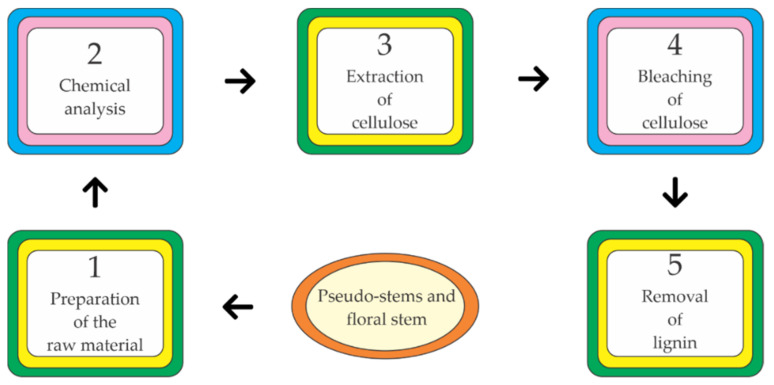
Cellulose extraction from the banana pseudostem and flower stem.

**Figure 10 molecules-26-05282-f010:**
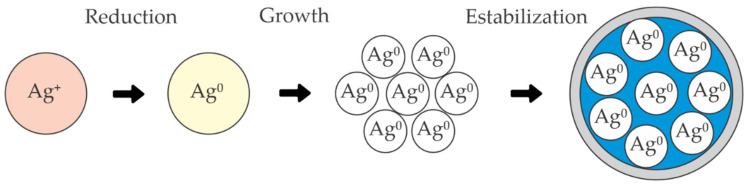
Scheme of the green synthesis of silver nanoparticles. Adapted from [166].

**Figure 11 molecules-26-05282-f011:**
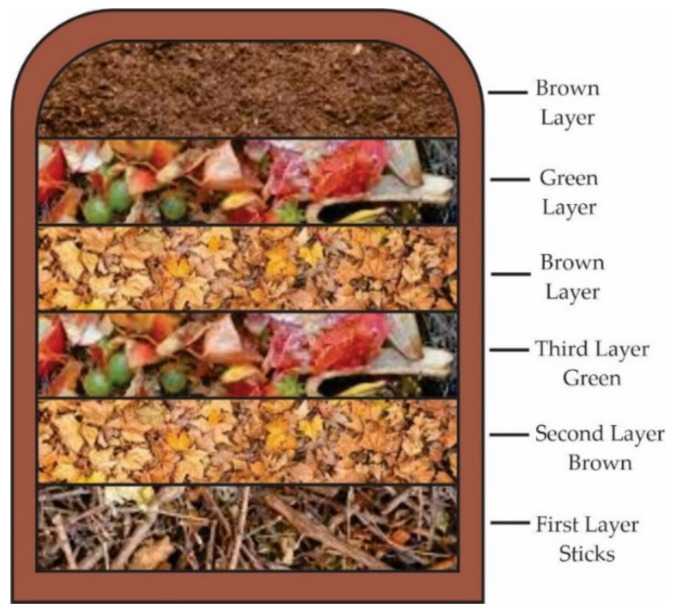
The layering of the compost pile.

**Table 1 molecules-26-05282-t001:** Classification of banana varieties based on morphological differences.

Producing Country	Variety	Genotype	Subgroup	References
Ecuador	Williams	AAA	Gros Michel	[34,35]
	Dwarf Cavendish	AAA	Cavendish	
	Maqueño	AAB	Red	
	Gros Michel Highgate	AAA	Gros Michel	
Philippines	Saba	ABB	Saba	[36,37]
	Cavendish	AAA	Cavendish	
	Lakatan	AAA	Lakatan	
Colombia	Banana Valery	AAA	Cavendish	[38]
	Hartón Enano	AAB	Plantain	
	Bocadillo del Quindío	AA	Sucrier	
	Guineo Común	AAAae	Lujujira	
India	Robusta	AAA	Cavendish	[39,40]
	Palayankodan	AAB	Mysore	
	Nendran	AAB	Cavendish	
	Red banana	AAA	Red	
Perú	Williams	AAA	Cavendish	[41]
	FHIA-17	AAAA	Non-defined	
	Cavendish Valery	AAA	Cavendish	

**Table 3 molecules-26-05282-t003:** Chemical composition of dry-based banana peel in solid state [66,67,68].

Components	(mg/100g Dry Peel)
Starch	0.78
Raw fiber	11.95
Crude protein	4.77
Calcium	0.36
Phosphorus	0.23
Lipids	1.15
Zinc	0.17
Ash	1.71

**Table 4 molecules-26-05282-t004:** Chemical composition of pseudostems from banana on a dry basis.

Components	(mg/100g Dry Peel)
Ash	28.3
Coal	38.3
Hydrogen	3.88
Sulfur	0.58
Lignin	5.2
Cellulose	35.3
hemicellulose	24.9

**Table 5 molecules-26-05282-t005:** Chemical composition of banana pulp on a dry basis [74,75].

Components	Composition (% DW)
Starch	18.4
Protein	3.1
Cellulose	0.8
Fat	0.62
Sugars	2.1
Ash	0.53
Phosphorus	0.13
Soluble carbohydrates	67.2
Ethereal extract	0.9

DW: Dry weight.

**Table 6 molecules-26-05282-t006:** Biofuel production studies using banana waste-loss.

Waste-Loss	Biofuel	Study Scale	Objective	Conclusions	References
Industrial waste-loss, peel and rachis of *Cavendish banana*	Bioethanol	Laboratory	Three experiments were carried out on 60 % ground banana peel. Enzymatic hydrolysis was carried out with conidia of the *Trichoderma viride* fungus and its subsequent alcoholic fermentation was with commercial dry active yeast *Saccharomyces cerevisiae*.	The results show that bioethanol can be obtained from the lignocellulosic waste-loss of the mature banana shell with a yield of 7% (*v*/*v*).	[83]
			Different mixtures of water and banana peels left to ferment for several days. The rachis was subjected to an alkaline hydrolysis process with NaOH to obtain ethanol.	Ethanol obtained were an average pH of 4.16, 3.75° Brix, with a concentration of 29 alcoholic degrees, with a distillate flow rate of 8.3 mL/s.	[84]
*Musa acuminata* Pseudo stems, rachis and peel	Biogas	Laboratory	In this study, the division into four stages of anaerobic digestion was carried out, in which anaerobic bioreactor and ripe banana peels were used.	After 8 h, 1 L of gas was obtained in the sampling system. A CO_2_ 99.97% concentration was obtained.	[86]
		Pilot plant	The biomass fraction was subjected to anaerobic digestion. Then a steam explosion pretreatment was implemented to increase the methane yield in the co-management of urban solid waste (pseudostem and rachis).	A yield of 363.29 L of CH_4_/kg of solid waste was obtained, equivalent to 56.32 kWh/tbh. The C/N ratio was adjusted to the 20–30 range.	[87]
Banana peel and stalk of *Musa* spp.	Biodiesel	Laboratory	The feasibility of using the ash derived from the peduncle of Musa spp as a highly effective renewable heterogeneous catalyst for the transesterification of underutilized non-edible *Ceiba pentandra* oil (CPO) was investigated.	Optimal process conditions were a catalyst concentration of 1.978% by weight, a reaction time of 60 min, with an expected yield of 99.36%, which was experimentally evaluated as 98.69 ± 0.18%.	[95]
			*Cryptococcus* sp. was used to produce oil with banana peel as raw material. Pretreatment of the banana peel with 1% sulfuric acid produced up to 4.5 g of glucose (11.2% yield) and 18.1 g of fructose (45.2% yield).	When *Cryptococcus* sp. was grown on the pretreated banana peel, its accumulated lipids up to 34.0%. The lipid had a high degree of monosaturated, which gave the resulting biodiesel a better quality.	[96]
Peel and rachis from *Musa paradisiaca*	Hydrogen	Laboratory	This study analyzes banana peels’ pretreatment effect in the photo-fermentative hydrogen production using brewery wastewater (BWW) in a discontinuous bioreactor.	The maximum hydrogen production yield (408.33 mL H_2_/L of wastewater) was achieved from the substrate, composed of 50% body weight pretreated with 1 g/L of banana peels for two hours and 50% medium standard.	[97]
			The drying process was carried out in an oven at 65 °C until a constant weight was achieved. The biomass pyrolysis was carried out in an electric oven. The system’s temperature was between 250 °C, 275 °C and 300 °C for 30 min in each experiment.	The results show that the pyrolysis process does not depend on the size of the particles or the content of reducing sugar. The production at 250 °C was about 1 mg of hydrogen. At 275 °C, production increased and then fell to 300 °C.	[98]
Banana peel *cavendish*	Biochar	Laboratory	This study focused on the yield of biochar and its production through the response surface methodology using a central compound design (CCD) in the pyrolysis of the batch reactor.	The biochar obtained by slow pyrolysis at 356.1 °C and heating rate of 14.7 °C/min had a bio-carbon yield and an O/C ratio of 58.8% 0.289, respectively.	[99]
			Five tests were carried out: Free (Control), 1% banana peel (P1), 2% banana peel (P2), 1% biochar (B1) and 2% biochar (B2) to evaluate the greenhouse gas emissions into the atmosphere.	Carbon dioxide emissions for treatments B1 and B2 decreased by 20% and 24% compared to banana peel, respectively.	[100]

**Table 7 molecules-26-05282-t007:** Studies on wastewater treatment using banana waste-loss.

Waste	Technique	Scale of Study	Study	Conclusions	References
Pseudo stems and shell of *Musa AAB*	Biofilters	Laboratory	The Cr (VI) was dispersed in 10 mL of acetone. The banana peels were washed with distilled water and subsequently dried at 120 °C for 12 h. Adsorption dynamics of the samples were analyzed utilizing batch reactors.	The bananas in the mixture removed almost all the Cr (VI) from the typical synthetic wastewater present in the tannery effluent (removal percentage of 93%).	[104]
			Cellulose was extracted from the pseudostems (39.12% cellulose and 72.71% holo-cellulose) to elaborate the biofilter. Efficiency was analyzed in terms of decreasing biological oxygen demand, chemical oxygen and suspended solids.	The biofilter was able to remove organic pollutants with efficiencies between 70.4 and 84.2%.	[118]
Rachis and shells from *Musa paradisiaca*	Activated carbon	Laboratory	The banana rachis was cut, dried and then burned to produce charcoal. Charcoal (3 g) was mixed with 75 mL of chromium-containing wastewater and then the chromium content in the filtrate was measured.	The removal efficiency was 99.8% at pH 6.7. The reduction efficiencies of biological, biochemical oxygen, chemical oxygen demand and chloride were 97%, 93% and 60%, respectively.	[107]
			Shell was dried under atmospheric conditions for two days, then dried in an oven at 80 °C for 18 h. HCl and NaOH were mixed. They were then smoothed and classified according to vibrating screens with the largest mesh size of 300 µm.	It was possible to reduce copper and chromium ion content by 55.5% and 61%. For dyes in textile waste, the average absorption capacity of the dye ion was 12.21% during 120 min.	[119]
*Musa* *Acuminata* shells	Coagulant	Laboratory	The shells were washed with distilled water and cut to 0.6 cm. They were then oven-dried for 48 h at 60 °C. Then, 0.5 g of the raw material was soaked in 100 mL of distilled water at room temperature and stirred at 120 rpm for one hour.	It could be concluded that the coagulant was highly effective in removing turbidity from synthetic wastewater with a removal efficiency of 88% under conditions of pH 1.0 and a dose of 100 mg/L.	[112]
*Musa* spp. shells	Biosorbent	Laboratory	Electrokinetic remediation was used for nickel removal. Different electrodes with different purge solutions (pH 3.5 and 7.0) and anionic surfactant (sodium dioctyl sulfosuccinate) were used. The shells were ground, washed and then dried in an oven for two days.	The removal efficiency was 74.8%. Biological remedies are considered an effective adsorbent material to prevent reverse osmosis flow, which provides a new idea for applying these products as an adsorption medium.	[111]
			The shell was washed with distilled water and ground to a particle size of 2 mm. Then, grounding and washing with n-hexane; subsequently, drying in an oven at 100 °C for 12 h.	The banana peel was an excellent absorbent for oil removal from wastewater with greater than 96% removal efficiency.	[108]
	Ceramic membrane	Industrial plant	The shell was washed with distilled water and ground. Clay was mixed in a mold subjected to 88 MPa. After milling, the raw materials were dried at 110 °C for four hours and sieved at 150 µm.	Filtration from the tannery revealed the removal of contaminants. Turbidity, dye content, suspended solids and biological and chemical oxygen demand were also reduced.	[120]
*Musa* *Sapientun* Pseudo stem	Biosorption	Laboratory	The pseudostems were dried under sunlight before drying in an oven for 24 h at 105 °C. All the experiments were carried out with the batch method, varying the pH (2.0–10.0), amount of biosorbent (0.10,0.50,1,1.5,2,2 g) and contact time (15–1440 min).	Removals for oxygen demand of 88%, ammonia nitrogen of 84%, suspended solids of 83%, turbidity of 75%, the color of 67% and oil and fat of 68% were achieved. Maximum elimination was achieved in two hours.	[121]

**Table 8 molecules-26-05282-t008:** Manufacture of bioplastics from banana waste-loss.

Residue	Biopolymer	Product	Study	Conclusions	References
Banana leaves	*Musa* spp. fiber sheets	Biodegradable packaging	Study on an industrial scale. Metal molds were elaborated to do the press and ironing of the sheets of banana and rice to give it shape and firmness. Rice powder was mixed with water and heated to a paste that was added to the banana sheets.	Two types of packaging were obtained (rectangular and cylindrical shape), capable of biodegrading 90% in approximately six months.	[136]
Pseudo stems	*Musa* *Acuminata* cellulose	Bioplastic	Laboratory-scale study. An enzymatic method was used to degrade lignin for 48 h at 28 °C with *Phanerochaete chrysosporium* and sieved to a size of 1000 microns. The material was sterilized for 15 min at 120 °C to avoid contamination.	The mechanical properties were 1.10 ± 0.15 Mpa of tension strength, 27.99 ± 14.72% elongation and 5.26 ± 1.46 MPa of elasticity, with complete biodegradability in approximately three months.	[124]
Pseudo stems	*Musa paradisiaca* fiber	Biodegradable utensils	Laboratory-scale study. The fibers extracted from the pseudostems were subjected to drying at 50 °C for seven hours. Purification and bleaching were carried out with 30 % NaOH, NaClO and water to harden them. They were then pulverized in a ball mill.	The average net weight of the plates obtained was 8.5 × 10^−3^ kg, a plate dimension of 156 mm × 14 mm. The average net weight of the vessels was 3.3 × 10^−3^ kg, translated into vessel dimensions of 69 mm × 81 mm with a capacity of 175 mL.	[138]
Banana shells	*Musa* *Paradisiaca* Starch	Bioplastic	Laboratory-scale study. Starch was extracted from the shells and mixed with HCl (0.5 M) and glycerol. Subsequently, NaOH 0.5 N was added and spread on a ceramic tile then placed in the oven at 120 °C.	The bioplastic film can support a weight close to 2 kg with enough traction and force. The manufactured bioplastic can be used as packaging material or as transport.	[128]
		Polyhydroxyalkanoate (PHA)	Laboratory-scale study. Banana starch was added to potential PHA producers such as *Staphylococcus aureus*, *Geobacillus stearothermophilus*, *Bacillus subtilis* and *Bacillus siamensis* during different incubation times.	*Geobacillus stearothermophilus* accumulated 84.63% PHA in 96 h. *Bacillus subtilis* accumulated 71.78% PHA in 24 h of incubation. *Bacillus siamensis* accumulated 77.55% and *Staphylococcus aureus* about 70.02% PHA in 24 h of incubation.	[139]
		Biodegradable planting bag	The shells were cut and oven-dried at 70 °C, then ground to 23 mm particle size. Subsequently, they were macerated and transformed into thermoplastic starch using eight different mixing concentrations.	The biodegradable plastic degraded quite quickly, with an average percentage weight loss of 65.1% within eight weeks.	[140]
Banana shells	*Musa* *paradisiaca* *fomatypica* pectin	Biodegradable plastic films	The pectin was dissolved in distilled water and five grams of shell extract was added and heated to 60 °C. The pectin solution was then mixed with chitosan and heated to 80 °C for 10 min. The mixture was placed in a mold and dried at 50–60 °C.	The bioplastic had a film thickness of 38.7 µm, water-resistance of 63.63 %, a tensile strength of 10.562 MPa and an elongation value of 58.33 %.	[141]
Banana pseudo stems	*Musa* spp. cellulose	Biodegradable film	The fibers were washed with an H_2_SO_4_ solution (2 g/L), filtered and mixed with NaOH (200 g/L) for 30 min. They were then heated to 80–90 °C, stirring at 200 rpm for four hours to form a cellulose film.	The cellulose film decomposed in the soil in four weeks, indicating excellent biodegradability compared to polystyrene (PE) plastic films.	[142]

## Data Availability

Not applicable.

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
