# Peer review of "Recovery of Banana Waste-Loss from Production and Processing: A Contribution to a Circular Economy"

_molecules, 2021, doi:10.3390/molecules26175282_

Round 1

Reviewer 1 Report

In general, the paper is very long containing non useful information. The aim of the paper should mentioned in the introduction and perhaps more precise to reduce the length of the paper. The link between the paper and the circular economy is not very clear. At least, the conclusion should make this link.  

Introduction: How many kg or tons in one "box"

Introduction: please mentioned the aim or objective of the paper

The text before subsection 2c is not useful.

section 2c: move in introduction section

Section 3: the text before 3i should be synthetized, a lot of non useful information

Table 7 and 8 are very interesting

conclusion: be sure that the conclusion is linked to the objective of the paper. Sentence like "thanks to" should be changed

Please use 2.1, 2.2 etc. instead of a, b, c.

Please use 3.1, 3.2 etc. instead of i, ii.

Author Response

All the authors of the manuscript are immensely grateful for the evaluators' contribution to improving the article. All the corrections are in the attached file, point by point, according to the evaluators' suggestions.

Reviewer 2 Report

This paper summarize the utilization of banana biomass and is important in the related filed. The authors should take into account the following comments before publication.

1. The author introduced the use of banana waste-loss in biofuel production, wastewater treatment, bioplatics, nanotechnology, organic fertilizers, but it seems that the authors spent too much writing on introducing related application instead of introducing how banana wastes are used in the related applications. For example, in b. Wastewater treatment, the authors spent a lot of writing in introducing different polluting sources, but only spent a few of them introducing how banana wastes were used in treating those pollutions. Another example, in d Nanotechnology, the authors spent a lot writing in introducing carbon nanotubes and nano-fertilizer, but failed to link them with banana wastes.

2. When introducing the use of banana wastes in different applications, personal discussion/opinion from the authors should added. For example, if certain types of banana wastes are suitable or not suitable in specific applications and why? The reasons related to specific chemical composition, physical structure of certain types of banana wastes should be added.

3. The section of d. Nanotechnology may be divided into several sections. The synthesis of silver nanoparticles is due to the reductive properties of polysaccharides from banana wastes. However, the reductive properties of banans wastes is the reason for the transformation of silver ion to silver (0), but is not the reason for the forming of nano-sized silver particle. So it cannot say synthesize nanoparticle by banana waste, but instead, it should say banana waste is reductive and can be used in organic synthesis. If banana wastes indeed assist the forming of nano-sized metal particles, the authors should discuss the related mechanisms,

What is the purpose of discussing carbon nano tubes? Do they have something to do with banana wastes?

The productions of bacterial cellulose and nano-cellulose from banana wastes are different. To produce bacterial cellulose, banana is used as the energy source to produce the bacterial that secretes bacterial cellulose. For produce nano-cellulose, it is to nano-size the cellulose in banana wastes. The author should specify the differences.

In sum, section d. Nanotechnology should be reformed in a more logical order.

4. I understand the author put very detail information in the Table, but the vital content should also be discussed in the main text. For example, in d. Nanotechnolgy, in the main text, the authors fail to discuss the applications of banana wastes in nano-fertilizer and production of BC, which makes the reader unable to understand what it is all about.

Author Response

All the authors of the manuscript are immensely grateful for the evaluators' contribution to improving the article. According to the evaluators ' suggestions, all the corrections are in the attached file, point by point.

Round 2

Reviewer 2 Report

After reviewing the revised manuscripr, I think the manuscript is greatly improved, but still needs minor revisions. The specific comments are as below:

The manuscript is greatly improved, but still needs minor revision before publication.
(1) The length of the manuscript can be further reduced by shortenning some of the common knowledge description, for example (not limited to) the first two paragraphs of 4.3 Bioplastics;
(2) The reviewer still think the authors should distinguish between lignocellulosic-based nano-cellulose and bacterial nano-cellulose. For lignocellulosic-based nano-cellulose, the cellulosic part of the banana wastes is the direct raw materials for nano-cellulose. However, for bacterial nanocellulose, banana wastes are the feed/energy source/nutrition source for the bacteria that produce bacterial nanocellulose. Using banana wastes as the feed can produce yeast, enzyme, and other good bacteria for various applications. It is suggested that the authors can add a small section that introduces banana wastes as the feed for microorgnisms, in which banana wastes used to feed bacteria that produces nanocellulose can be included.

Author Response

All the manuscript's authors thank the excellent reviews that help to improve the manuscript's quality. All the corrections are in the document attached. 
